# Effects of Nitrogen Dioxide on Biochemical Responses in 41 Garden Plants

**DOI:** 10.3390/plants8020045

**Published:** 2019-02-16

**Authors:** Qianqian Sheng, Zunling Zhu

**Affiliations:** 1College of Landscape Architecture, Nanjing Forestry University, Nanjing 210037, China; qqs@njfu.edu.cn; 2College of Art & Design, Nanjing Forestry University, Nanjing 210037, China

**Keywords:** garden plants, chlorophyll, POD, soluble protein, MDA, mineral ions

## Abstract

Nitrogen dioxide (NO_2_) at a high concentration is among the most common and harmful air pollutants. The present study aimed to explore the physiological responses of plants exposed to NO_2_. A total of 41 plants were classified into 13 functional groups according to the Angiosperm Phylogeny Group classification system. The plants were exposed to 6 μL/L NO_2_ in an open-top glass chamber. The physiological parameters (chlorophyll (Chl) content, peroxidase (POD) activity, and soluble protein and malondialdehyde (MDA) concentrations) and leaf mineral ion contents (nitrogen (N^+^), phosphorus (P^+^), potassium (K^+^), calcium (Ca^2+^), magnesium (Mg^2+^), manganese (Mn^2+^), and zinc (Zn^2+^)) of 41 garden plants were measured. After NO_2_ exposure, the plants were subsequently transferred to a natural environment for a 30-d recovery to determine whether they could recover naturally and resume normal growth. The results showed that NO_2_ polluted the plants and that NO_2_ exposure affected leaf Chl contents in most functional groups. Increases in both POD activity and soluble protein and MDA concentrations as well as changes in mineral ion concentrations could act as signals for inducing defense responses. Furthermore, antioxidant status played an important role in plant protection against NO_2_-induced oxidative damage. NO_2_ poses a pollution risk to plant systems, and antioxidant status plays an important role in plant protection against NO_2_-induced oxidative damage. In conditions of strong air pollution, more evergreen plants may be considered in landscape design, particularly in seasonal regions. The results of this study may provide useful data for the selection of landscaping plants in NO_2_ polluted areas.

## 1. Introduction

The emission of atmospheric nitrogen dioxide (NO_2_) has been a growing trend in many areas worldwide during the last few decades [1], especially in some Asian countries [2]. Due to the development of industrialized production and the continuous increase in automobile exhaust emissions, the NO_2_ concentration will likely continue to increase, and the NO_2_ level will continue to exceed the standard level of NO_2_ pollution [3]. 

Nitrogen dioxide (NO_2_) is a precursor of harmful secondary air pollutants such as ozone and particulate matter [4,5,6]. The use of ecological methods to reduce air concentrations, such as plant absorption and catabolism of atmospheric NO_2_, is very important. At present, scholars hold different views towards the effects of NO_2_ on plants. One hypothesis purports that, by being metabolized and incorporated in the nitrate assimilation pathway, NO_2_ can form organic nitrogenous compounds in plants [7]; this process does not injure leaves [8]. Another hypothesis purports that most plants exhibit both low amounts of NO_2_-N incorporation into total plant nitrogen (N) and resistance to NO_2_ [9]. Complex physiological responses can be triggered when plants are exposed to NO_2_, including changes in antioxidant enzyme activity [10,11], N metabolic enzyme activity [12,13], and both the components and distribution of nitrogenous metabolic products in plant tissues [14]. 

The dissolution of low concentrations of NO_2_ in water can result in the formation of nitrate and nitrite, both of which are used by plants during the normal process of nitrate metabolism; as such, NO_2_ may act as an airborne fertilizer [15]. However, high concentrations of NO_2_ can lead to excessive accumulations of nitrite (NO_2_^−^) [16] and cell acidification [17], which lead to negative responses such as the generation of reactive oxygen species (ROS) and inhibition of both N assimilation and plant growth, further causing acute damage to leaves, whole-plant chlorosis or even death. However, exposure of NO_2_ to different plant species elicits different physiological responses. Therefore, the effects of NO_2_ exposure on plants remain highly controversial, and a unified conclusion has not been reached. In addition, information concerning different plants that are highly tolerant to NO_2_ and their natural recovery is scarce.

To identify garden plants that exhibit good absorption and strong resistance, we studied the physiological responses of 41 garden plants exposed to specific NO_2_ environments under controlled conditions in the laboratory; the plants were located in Jiangsu Province. Few studies have investigated whether these 41 plants species, which are commonly planted along roadsides in urban areas of many countries, have high NO_2_ absorption capacity and/or are resistant to the effects of NO_2_. Several studies have focused on the effects of NO_2_ concentration on plant growth and reported both that low concentrations (0.1 µl/L) of NO_2_ did not significantly influence the height, leaf area or dry weight of 1-year-old *Buxus sinica* seedlings [18] and that 0.5 µl/L NO_2_ significantly stimulated the leaf growth of *Populus deltoides* and *Populus nigra Italica*; however, a higher NO_2_ concentration (1 µl/L) significantly reduced the stem growth. When *Arabidopsis thaliana* plants were exposed to 0.85, 2, 4.25, and 9.4 µl/L NO_2_, the plants exhibited acute visible leaf injuries, and the plants eventually died. When those plants were treated with NO_2_ concentrations between 2 and 4.25 µl/L, some injury occurred, but the plants remained alive; furthermore, no significant differences were recorded between leaf growth and chlorophyll (Chl) content. Therefore, we chose 6 µl/L as the NO_2_ fumigation concentration. 

The main objectives of our study were to (1) evaluate the physiological responses of 41 plants exposed to simulated NO_2_ pollution; (2) determine the leaf mineral ion contents as a function of the atmospheric concentrations of NO_2_, ultimately reflecting the NO_2_ pollution associated with these different kinds of plant life forms; (3) identify plant life forms that have a higher NO_2_ absorptive capacity than do other plant life forms; and (4) evaluate the natural recovery of plants following NO_2_ exposure to ensure whether they can adjust to normal NO_2_ levels. The results will be used for selecting landscaping plants to improve polluted and potentially polluted areas.

## 2. Materials and Methods 

### 2.1. Plant Materials and Growth Conditions

Plant species were selected using pro rata sampling and ordinal classification according to the Angiosperm Phylogeny Group [19], i.e., species were sampled according to the proportion of the number of species within each order [20]. In the present study, 41 plant species distributed mostly among 28 families and 22 orders were selected. The experiment was conducted in the garden plant laboratory and greenhouse of Nanjing Forestry University in Nanjing, Jiangsu Province, China (latitude 32.07°N, longitude 118.8°E). Table 1 lists the detailed information of these 41 plant species with respect to their plant functional group classification. 

From March to June 2017, the seedlings of 41 plant species were collected from the mountains in Changzhou, Jiangsu Province, China. 

The altitude of the sample collection zone ranged from 100 to 600 m, and the local soil consisted of mostly yellow brown soil. The soil contained alkali-hydrolyzable N, available P, available K, and organic matters at 47.05 mg/kg, 5.91 mg/kg, 145.93 mg/kg, and 10.35 g/kg, respectively, with a pH value of 6.51. The collections involved a minimum of 10 mature leaves and 1–2-year-old tree and shrub seedlings whose height ranged from 10–30 cm. After collection, all the seedlings were immediately brought to the laboratory and transplanted into individual pots whose dimensions were 20 cm (diameter of open top) × 15 cm (height) × 30 cm (diameter of flat bottom); the pots were previously filled with vermiculite and peat (1:1, *v*/*v*), which were thoroughly mixed together. During the culture period, all seedlings were watered with half-strength Hoagland nutrient solution until new leaves emerged. Afterward, the seedlings were watered with tap water for 3 d; this rate was in accordance with the water evaporation rate of the soil as described by Allen et al. [21]. Each pot was also irrigated with 1 L of full-strength Hoagland nutrient solution biweekly. The plants were grown for at least 2 months prior to starting NO_2_ treatments and measuring plant physiological responses. The plants were grown in an artificially controlled greenhouse: the air temperature was 25–28 °C; the relative humidity was 60–70%; and the photoperiod was 14 h, produced via 500–900 μmol photons/m^2^/s of photosynthetically active radiation (PAR). 

### 2.2. NO_2_ Treatment

Open-top glass chambers (50 cm × 50 cm × 50 cm) were constructed for the NO_2_ fumigation applications. Considering that plants remained alive in the condition of NO_2_ below 4.25 μL/L [22], but died when 9.4 μL/L NO_2_ is used [9], the NO_2_ concentration was determined at 6 μL/L in this experiment. NO_2_ gas was supplied directly from cylinders (400 μL/L NO_2_; the velocity of gas flow was 1 L/min, which was controlled by a gas flow meter). Seedlings of the control group were placed in another climate chamber, which was quantitatively flushed with filtered air (free of NO_2_) at the same time. The climate chamber was subjected to a photoperiod of 13 h, a temperature of 25/20 ± 3 °C (day/night), and a relative humidity of 60/50 ± 4% (day/night). The control and NO_2_-treated seedlings (10 replications per each treatment) were fumigated for 3 d at 6 h/d.

The collected leaves were exposed to 100 L/min air. NO_2_ concentrations within the climate chamber were monitored by an NO_2_ analyzer (model ML Series), and NO_2_ concentrations were recorded every 1 h. All the recorded data were downloaded via a micro-USB connection onto a PC using specific software (CairSoft^®^). 

Physiological indices (Chl a content, Chl b content, carotenoids, total Chl content, peroxidase (POD) activity, soluble protein concentration, and malondialdehyde (MDA) concentration) and mineral ion contents (nitrogen (N^+^), phosphorus (P^+^), potassium (K^+^), calcium (Ca^2+^), magnesium (Mg^2+^), manganese (Mn^2+^), and zinc (Zn^2+^)) in the leaves were measured upon exposure to NO_2_ for 0 h and 72 h and after 30 d of natural recovery (NR). The mean leaf area, Chl content, POD activity, MDA concentration and soluble protein concentration were calculated based on the life form (e.g., herb, shrub, or tree), leaf type (e.g., needle-like or broadleaf), phylogeny (e.g., gymnosperm or angiosperm), and photosynthetic pathway (e.g., C_3_ or C_4_). Furthermore, angiosperms consist of dicotyledons and monocotyledons [23,24]. Each treatment group included three replicate samples. All experimental groups and control groups were placed separately in several identical chambers.

### 2.3. Measurement of Chl Contents

Chl contents were determined by the spectrophotometric method (759S, Lengguang Tech). Briefly, fresh leaves and distilled water were mixed together, after which the mixture was pulverized; the homogenate was then extracted by 90% ethanol. The absorbance of the supernatant was measured at 663 and 645 nm using a spectrophotometer. The Chl content was expressed as milligrams per gram of fresh weight (fw).

### 2.4. Measurement of Lipid Peroxidation Levels

Lipid peroxidation levels are reflected by MDA concentrations. Lipid peroxidation can produce MDA, which serves as a sensitive diagnostic index for plants under oxidative injury [25]. Two milliliters of 5% trichloroacetic acid (*w*/*v*) were added to fresh leaf samples, which were then ground, and the resulting homogenate was subsequently centrifuged at 300 *g* for 10 min. Afterward, 2 ml of 0.67% thiobarbituric acid (*w*/*v*) was added to 2 ml of the resultant supernatant, after which the mixture was heated at 100 °C for 30 min and then centrifuged at 300 *g* for 10 min. The absorbance of the supernatant was subsequently measured at 450 nm, 532 nm, and 600 nm. The MDA content was then calculated based on the following formula:C/µg/Ml/L = 6.45 × (A_532_ − A_600_) − 0.56A_450_,
where A_450_, A_532_ and A_600_ represent the absorbance value at a wavelength of 450, 532, and 600 nm, respectively [26].

The lipid peroxidation levels were expressed as micromoles of MDA per gram of fw.

### 2.5. Antioxidant Enzyme Activity Assays

The guaiacol method was used to determine the POD activity. Phosphate buffer (0.05 mol/L, pH 5.5) was added to fresh leaf samples (0.2 g), which were then ground, and the resulting homogenate was subsequently centrifuged at 300 *g* for 10 min at 4 °C. The supernatant was then added to a reaction mixture consisting of 2.9 ml of 0.05 mol/L phosphate buffer, 1 mL of 0.05 mol/L guaiacol (the donor) and 1 mL of 2% H_2_O_2_ (the substrate) for a 15-min reaction. The absorbance at 470 nm was measured, and the activity was expressed as the optical density at 470 nm per minute per gram of fw, based on the following formula:
POD activity [u/(g·min)] = (△A_470_ × *V_t_*)/(*W* × *V_S_* × 0.01 × *t*)
where △A_470_ is the variation of the absorbance within the reaction time period, *W* is the fresh weight of the sample (g), *t* is the reaction time (1 min), *V_t_* is the total volume of enzyme extract (6 mL), and *V_s_* is the amount of use of the extract liquid at the time of measurement (0.1 mL). 

### 2.6. Determination of Soluble Protein

Frozen total leaf tissue (third and fourth leaves) immersed in liquid N was ground to a fine powder using a mortar and pestle. Samples (1 g) were then extracted with 8 ml of distilled water and incubated for 1 h, after which the extracts were centrifuged at 500 *g* for 20 min at 4 °C. The amount of soluble protein was subsequently determined by the Bradford method using bovine serum albumin [27,28]. The soluble protein concentrations were expressed as milligrams per gram of fw.

### 2.7. Determination of Mineral Ions

After the plants were fumigated, 30 g of leaf material was removed and then dried in an oven at 105 °C for 30 min, after which it was dried at 80 °C to constant weight. The material was subsequently passed through a 40-mesh sieve, and then boiled in accordance with the nitric acid-perchloric acid method. N^+^ was determined by colorimetry; K^+^, by flame spectrophotometry; P^+^, by molybdenum antimony spectrophotometry; Ca^2+^, Mg^2+^, Mn^2+^, and Zn^2+^, by atomic spectrophotometry [29].

### 2.8. Statistical Analysis

Measurement data are transformed to folds between the measurements in the experimental groups and those in the corresponding –NO_2_ (normal control) group. They are presented as the means ± SD (n = 3) and processed with SPSS 24.0 (IBM; Armonk, NY, USA). Levene’s test was performed to determine the homogeneity of variances, followed by one-way analysis of variance (ANOVA) for comparisons among the different treatment groups in the same functional group. In case of heterogeneity, nonparametric Kruskal-Wallis one-factor analysis of variance was used. Bonferroni corrections were performed. Values of * *p* < 0.05 and ** *p* < 0.01 were considered statistically significant and highly significant, respectively.

## 3. Results

### 3.1. Variations in Leaf Area and Physiological Responses across Different Functional Groups in Response to NO_2_ Treatments

The values of the various leaf characteristics differed greatly between functional groups (Table 2). The smallest variation in the Chl a content was observed in evergreens (0.9 ± 0.05 fold that before NO_2_ exposure; *p* < 0.05); the greatest, in conifers (0.31 ± 0.11 fold that before NO_2_ exposure; *p* < 0.01). The variations in Chl b content ranged from 0.94 ± 0.09 fold in evergreens (*p* > 0.05) to 0.35 ± 0.04 fold in conifers (*p* < 0.01). The variations in the mean carotenoid content ranged from 0.92 ± 0.17 fold in evergreens (*p* > 0.05) to 0.24 ± 0.02 fold in conifers (*p* < 0.01). 

Under NO_2_ stress, a very significant decrease in Chl content occurred and the relative Chl content significantly decreased to 0.81 ± 0.47 fold that of the control (*p* < 0.01). After recovery, this content greatly increased. The results are summarized in Table 3.

### 3.2. Effects of NO_2_ Treatment on Various Leaf Traits

As shown in Table 3, compared with the control treatment (without NO_2_), the NO_2_ treatment led to a significant increase in POD activity, soluble protein and MDA concentrations; however, the activity and concentrations decreased following natural recovery for 30 d. 

The relative POD activity increased to 3.74 ± 0.68 folds that of the control (*p* < 0.01) in response to NO_2_ treatment but decreased to 1.68 ± 0.95 folds after 30 d of recovery, and the relative soluble protein concentration increased to 1.07 ± 0.70 folds that of the control (*p* < 0.01) in response to NO_2_ treatment, but decreased to 0.73 ± 0.42 fold after 30 d of recovery. The relative MDA concentration increased to 3.48 ± 7.49 folds that of the control (*p* < 0.01) in response to NO_2_ treatment, but decreased to 1.80 ± 2.53 folds after 30 d of recovery. Interestingly, the relative POD activity and the soluble protein and MDA concentrations in evergreens subjected to the NO_2_ and NR treatments were noticeably lower than those in the other functional groups subjected to the same treatments. 

### 3.3. Macro- and Microelements in the Control, NO_2_-Treated and Naturally Recovered Plants

For normal leaves, the weight percent of chemical elements was dominated by the macroelements (N, P, Ca, K, and Mg), which accounted for approximately 85%; the microelements (Mn and Zn) constituted a low weight percent. 

The weight percent of the chemical elements was significantly affected by NO_2_ treatment (Table 4 and Table 5). Five macroelements (N, P, K, Ca and Mg) were detected in the control, NO_2_-treated, and recovered leaves. Compared with that in the control treatment, the content of N, K, Ca, Mg, and Mn in the NO_2_ treatment increased. Significant differences in the weight percent of N occurred between the control, NO_2_-treated leaves and naturally recovered leaves (*p* < 0.05). Interestingly, compared with those in evergreens in the control treatment, the N, K, and Ca contents in evergreens in the NO_2_ treatment tended to increase; their rates of increase (1.20 ± 0.41, 1.53 ± 0.79, and 1.94 ± 0.54 folds, respectively) were the highest among the different functional groups. 

Compared with those in evergreens in the control treatment, the N, K and Ca contents in evergreens in the NR treatment tended to increase; their rates of increase (1.31 ± 0.33, 1.94 ± 0.74 and 1.09 ± 0.24 folds, respectively) were noticeably higher than those in the other functional groups. 

## 4. Discussion

Air pollutants cause damage to plants in two forms: chronic injury, due to a long-term exposure to pollutants at a low concentration, and acute injury, as a consequence of a short-term fumigation with pollutants at a high concentration [15]. This study focused on the latter damage caused by a short-term fumigation with NO_2_ at 6 μL/L. To study plant capacity for NO_2_ absorption, we investigated plant physiological responses such as changes in Chl content, lipid peroxidation, antioxidant enzyme activity, and soluble protein concentration. Atmospheric NO_2_ enters a leaf mainly through stomata, and a small amount of this NO_2_ can be assimilated into organic compounds [30]. Exposure to various concentrations of NO_2_ differentially affected leaf Chl contents across functional groups. In the present study, reduced Chl contents might result from a damaged defense system and, consequently, unbalanced metabolism.

Under environmental stress, membrane lipids and proteins are the accessible targets of ROS in plants [31,32]. Membrane lipids and proteins are considered reliable indicators of the controlled modulation of ROS levels and oxidative stress [33]. Therefore, to further study lipid peroxidation and protein oxidation, we measured MDA and soluble protein contents. MDA, which is a product of lipid peroxidation, and electrolyte leakage allow the assessment of cell membrane injury. In the various functional groups, the MDA and soluble protein concentrations increased, and oxidative stress occurred in response to exposure to 6 µl/L NO_2_. These findings suggest that NO_2_ stimulates defense reactions, and these defense reactions might counteract ROS that are generated via elevated antioxidant defense systems in affected tissues; after NR and NO_2_ exposure, the production of ROS decreased, and the capacity of the antioxidant system was restored. As a result, the plants experienced substantial oxidative stress, but they recovered and resumed normal growth. Zhang et al. [26] investigated the enzymatic browning and antioxidant activities in harvested litchi (*Litchi chinensis* Sonn.) fruit after exposure to apple polyphenols, and proposed that plants could produce reactive oxygen species (ROS) under salt stress and that plants had their own ROS scavenging systems (antioxidant systems). Our study showed that plants built their ROS scavenging systems by resorting to activity metabolism.

Additionally, the activity of antioxidant enzymes (e.g., POD) was tested under the same treatment conditions. NO_2_ increased the activity of POD, which might be associated with increased ROS generation (alternatively, since NO_2_ reduces chlorophyll, fewer ROS are produced from the electron transport change leakage; the reduced ROS burden leads to greater activity of POD). This result suggested that the induced antioxidant defenses are regulated by an ROS-mediated signaling pathway. POD is the primary H_2_O_2_-scavenging enzyme in plant cells, and POD has a high affinity for H_2_O_2_ [34]. In this study, NO_2_ markedly increased the POD activity. After NO_2_ exposure, POD activity significantly increased and eliminated a portion of the generated H_2_O_2_; POD was the most efficient scavenging enzyme at decreasing the cellular levels of H_2_O_2_ in plant cells under NO_2_ stress. Moreover, increased POD activity can contribute to overall cellular resistance to NO_2_ stress, as POD both participates in many other cell processes involved in plant defense reactions [35] and can initiate cell wall-toughening events such as phenolic cross-linking and lignification, which can strengthen leaf and stem tissues against potential damage [36]. POD plays a role both in cell wall toughening and in the production of secondary metabolites; these roles, as well as its concurrent oxidant and antioxidant capabilities, make POD an important factor in the integrated defense response of plants to NO_2_ stress. According to a similar study conducted by Fang et al. [11], NO_2_ exposure elevated the levels of lipid peroxidation and protein oxidation, accompanied by the introduction of antioxidant enzyme activities. The results of our study were consistent with those reported [11].

The balance of plant ions reflects the degree of environmental stability; it is the premise of maintaining normal physiological activities inside plant cells. Adversities can affect the normal physiology of plants: they alter ion concentrations in plant organs and thus disrupt the dynamic balance among different ions. Although increased NO_2_ stimulates plant growth, exposure to a high concentration of NO_2_ within a short period reduces mineral ion concentrations in plant issues [37,38]. Dilution effects especially occur in the leaves, reducing plant photosynthesis and inhibiting plant growth [39]. In the present study, differences in ion concentrations were observed among the different functional groups under NO_2_ exposure; the ions most clearly affected included K^+^, Ca^2+^, Mn^2+^, and Zn^2+^. N^+^, K^+^, and Ca^2+^ accumulated more in the evergreen group than in the other functional groups. In addition, no significant effect on Mg^2+^ concentrations occurred in response to the NO_2_ or NR treatment. N^+^ and Mg^2+^ are major components involved in the synthesis of Chl, which is required for photosynthesis. We believe that NO_2_ exposure is conducive to the synthesis of organic pigments and promotes photosynthesis. The variation in Mg^2+^ concentrations shows that evergreens can maintain cellular stability under NO_2_ exposure and exhibit specific resistance to NO_2_. P^+^ can promote protein synthesis; however, increasing concentrations of soluble protein can reduce the P^+^ content. Ca^2+^ is a major component of the cell wall and is involved in maintaining the stability of cell membrane structure. Mineral ion concentrations differed under NO_2_ exposure. This result was similar to those of previous studies investigating mineral ion chemical properties: a certain degree of synergistic or antagonistic effect resulted from the process of plant absorbing and transferring ions, and a limited number of cells contain metal ions [40].

## 5. Conclusions

NO_2_ exposure affected leaf Chl levels. Furthermore, NO_2_ increased the levels of lipid peroxidation and protein dissolution, which was accompanied by the induction of POD activity and a change in antioxidant content. Lastly, plants injured from NO_2_ exposure could recover and resume normal growth.

We also studied plant mineral ion concentrations under experimental conditions and confirmed that these concentrations change in response to NO_2_ stress. N, an essential plant macronutrient, is a major limiting factor regulating plant growth and development [41]. Changes in both physiological responses and mineral ion concentrations induced by NO_2_ exposure significantly affect plant growth. Similarly, in the present study [42], compared with those in other functional groups, species in some functional groups, especially evergreens, exhibited higher leaf area and leaf N levels, resulting in markedly lower Chl contents, POD activity, soluble protein, and MDA concentrations. 

Our results imply that NO_2_ constitutes a pollution risk to plant systems and that antioxidant status plays an important role in plant protection against NO_2_-induced oxidative damage. However, plants can naturally recover from this damage.

## Figures and Tables

**Table 1 plants-08-00045-t001:** 41 plant species used in this study and their corresponding Angiosperm Phylogeny Group (APG) (APG, 1998) and plant functional group classification.

NO.	Order	Family	Genus	Species	Functional Group			N
					Life Form	Leaf Type	Phylogeny	
1	Arales	Araceae	*Acorus*	*tatarinowii*	C_3_ herb	Broadleaf	Angiosperm. Monocotyledon	8
2	Asterales	Asteraceae	*Farfugium*	*japonicum*	C_3_ herb	Broadleaf	AngiospermDicotyledon	8
3	Celastrales	Celastraceae	*Buxus*	*megistophylla*	Shrub	Broadleaf	Angiosperm Dicotyledon	8
4	Celastrales	Celastraceae	*Buxus*	*megistophylla*	Shrub	Broadleaf	Angiosperm Dicotyledon	8
5	Coniferales	Cupressaceae	*Sabina*	*procumbens*	Evergreen tree	Needle-like	Gymnosperm	8
6	Contortae	Apocynaceae	*Vinca*	*major*	Shrub	Broadleaf	Angiosperm Dicotyledon	8
7	Contortae	*Oleaceae*	*Ligustrum*	*japonicum* ’Howardii’	Shrub	Broadleaf	Angiosperm Dicotyledon	8
8	Contortae	*Oleaceae*	*Jasminum*	*mesnyi*	Vine Shrub	Broadleaf	Angiosperm Dicotyledon	8
9	Contortae	Oleaceae	*Osmanthus*	*fragrans* ‘Boyejingui’	Evergreen tree	Broadleaf	Angiosperm Dicotyledon	8
10	Cornales	Cornaceae	*Aucuba*	*japonica* ‘Variegata’	Shrub	Broadleaf	Angiosperm Dicotyledon	8
11	Dipsacales	Caprifoliaceae	*Viburnum*	*odoratissimum*	Shrub	Broadleaf	Angiosperm Dicotyledon	8
12	Ebenales	Symplocaceae	*Symplocos*	*tetragona*	Evergreen tree	Broadleaf	Angiosperm Dicotyledon	8
13	Ericales	Ericaceae	*Rhododendron*	*pulchrum*	Shrub	Broadleaf	Angiosperm Dicotyledon	8
14	Fagales	Betulaceae	*Carpinus*	*putoensis*	Deciduous tree	Broadleaf	Angiosperm Dicotyledon	8
15	Fagales	Betulaceae	*Carpinus*	*betulus*	Deciduous tree	Broadleaf	Angiosperm Dicotyledon	8
16	Gentianales	Rubiaceae	*Serissa*	*japonica*	Shrub	Broadleaf	Angiosperm Dicotyledon	8
17	Ginkgoales	Ginkgoaceae	*Ginkgo*	*biloba*	Deciduous tree	Broadleaf	Gymnosperm	8
18	Ginkgoales	Ginkgoaceae	*Ginkgo*	*biloba* ‘Wannianjin’	Deciduous tree	Broadleaf	Gymnosperm	8
19	Graminales	Gramineae	*Cynodon*	*dactylon*	C_4_ herb	Broadleaf	AngiospermMonocotyledon	8
20	Liliflorae	Liliaceae	*Hosta*	*plantaginea*	C_3_ herb	Broadleaf	Angiosperm Monocotyledon	8
21	Liliflorae	Liliaceae	*Ophiopogon*	*japonicus* var. *nana*	C_3_ herb	Broadleaf	Angiosperm Monocotyledon	8
22	MagnoliaIes	Calycanthaceae	*Calycanthus*	*chinensis*	Shrub	Broadleaf	Angiosperm Dicotyledon	8
23	*Pinales*	Pinaceae	*Cedrus*	*deodara*	Evergreen tree	Needle-like	Gymnosperm	
24	Poales	Poaceae	*Zoysia*	*japonica*	C_4_ herb	Broadleaf	Angiosperm Monocotyledon	8
25	Poales	Poaceae	*Carex*	*heterostachya*	C_4_ herb	Broadleaf	Angiosperm Monocotyledon	8
26	Poales	Poaceae	*Axonopus*	*compressus*	C_4_ herb	Broadleaf	Angiosperm Monocotyledon	8
27	Ranales	Lauraceae	*Lindera*	*glauca*	Deciduous tree	Broadleaf	Angiosperm Dicotyledon	8
28	Ranunculales	Berberidaceae	*Mahonia*	*fortunei*	Shrub	Broadleaf	Angiosperm Dicotyledon	8
29	Ranunculales	Berberidaceae	*Nandina*	*domestica*	Shrub	Broadleaf	Angiosperm Dicotyledon	8
30	Rosales	Pittosporaceae	*Pittosporum*	*tobira*	Shrub	Broadleaf	Angiosperm Dicotyledon	8
31	Rosales	Hamamelidaceae	*Loropetalum*	*chinense* var. *rubrum*	Shrub	Broadleaf	Angiosperm Dicotyledon	8
32	Rubiales	Rubiaceae	*Gardenia*	*jasminoides* Ellis var. grandiflora	Shrub	Broadleaf	Angiosperm Dicotyledon	8
33	Rubiales	Caprifoliaceae	*Weigela*	*florida* cv. Red Prince	Shrub	Broadleaf	Angiosperm Dicotyledon	8
34	Salicales	Salicaceae	*Salix*	*Integra* ‘Hakuro Nishiki’	Shrub	Broadleaf	Angiosperm Dicotyledon	8
35	Sapindales	Buxaceae	*Buxus*	*sinica*	Shrub	Broadleaf	Angiosperm Dicotyledon	8
36	Saxifragales	Saxifragaceae	*Hydrangea*	*macrophylla*	Shrub	Broadleaf	Angiosperm Dicotyledon	8
37	Scrophulariales	Verbenaceae	*Clerodendrum*	*trichotomum*	Shrub	Broadleaf	Angiosperm Dicotyledon	8
38	Umbelliflorae	Cornaceae	*Swida*	*alba*	Shrub	Broadleaf	Angiosperm Dicotyledon	8
39	Umbelliflorae	Araliaceae	*Fatshedera*	*lizei*	Shrub	Broadleaf	Angiosperm Dicotyledon	8
40	Umbelliflorae	Araliaceae	*Hedera*	*nepalensis* var. *sinensis*	Vine shrub	Broadleaf	Angiosperm Dicotyledon	8
41	Violales	Theaceae	*Camellia*	*sasanqua*	Evergreen tree	Broadleaf	Angiosperm Dicotyledon	8

**Table 2 plants-08-00045-t002:** Relative leaf chlorophyll (Chl) a content, Chl b content and carotenoid contents (folds) in different plant functional groups treated without nitrogen dioxide (NO_2_) (−NO_2_), with 72 h NO_2_ stress (+NO_2_) or after 30 d of natural recovery (NR).

	Number of Species	Specific Leaf Area (cm^2^/g Dry weight)	Relative Chl a Content			Relative Chl b Content			Relative Carotenoid Content		
			−NO_2_	+NO_2_	NR	−NO_2_	+NO_2_	NR	−NO_2_	+NO_2_	NR
**Life from**											
Herb	8	69,216	1	0.43 ± 0.06**	0.9 ± 0.09	1	0.61 ± 0.12**	0.83 ± 0.09	1	0.34 ± 0.17**	0.66 ± 0.30*
Shrub	23	111,837	1	0.58 ± 0.09**	0.96 ± 0.05	1	0.35 ± 0.04**	0.9 ± 0.14	1	0.48 ± 0.04**	0.96 ± 0.05
Tree	10	129,284	1	0.7 ± 0.07**	1.1 ± 0.06	1	0.88 ± 0.06	1.21 ± 0.12*	1	0.33 ± 0.03**	0.92 ± 0.03**
Evergreen	5	215,042	1	0.9 ± 0.05*	1.15 ± 0.07*	1	0.94 ± 0.09	1.25 ± 0.07**	1	0.92 ± 0.17	1.03 ± 0.04
Deciduous	5	17,669	1	0.59 ± 0.05**	1.07 ± 0.07	1	0.84 ± 0.16	1.19 ± 0.08	1	0.49 ± 0.03**	0.85 ± 0.01**
Broadleaf	39	126,381	1	0.62 ± 0.08**	0.95 ± 0.08	1	0.57 ± 0.14**	0.94 ± 0.05	1	0.47 ± 0.01**	0.86 ± 0.06**
Needle-like	2	987,404	1	0.31 ± 0.11**	0.82 ± 0.03*	1	0.35 ± 0.04**	0.75 ± 0.1**	1	0.24 ± 0.02**	0.51 ± 0.03**
**Phylogeny**											
Gymnosperm	4	469,349	1	0.39 ± 0.03**	0.92 ± 0.04*	1	0.48 ± 0.04**	0.73 ± 0.18*	1	0.28 ± 0.06**	0.9 ± 0.04*
Angiosperm	37	104,701	1	0.57 ± 0.06**	0.99 ± 0.08	1	0.7 ± 0.09*	0.93 ± 0.12	1	0.48 ± 0.04**	0.93 ± 0.01*
Monocotyledon	7	68,862	1	0.38 ± 0.07**	0.82 ± 0.07**	1	0.55 ± 0.05**	0.79 ± 0.09*	1	0.36 ± 0.02**	0.64 ± 0.03**
Dicotyledon	30	113,064	1	0.61 ± 0.02**	1.02 ± 0.1	1	0.73 ± 0.09*	0.96 ± 0.19	1	0.52 ± 0.02**	1.01 ± 0.02
**Photosynthetic pathway**											
C_4_ herb	4	2668	1	0.36 ± 0.03**	0.96 ± 0.08	1	0.58 ± 0.11**	0.9 ± 0.08	1	0.35 ± 0.01**	0.83 ± 0.01**
C_3_ herb	4	91,399	1	0.55 ± 0.04**	1.45 ± 0.1**	1	0.54 ± 0.21*	0.7 ± 0.24	1	0.33 ± 0.06**	0.52 ± 0.04**
All species	41	146,532	1	0.59 ± 0.03**	0.97 ± 0.08	1	0.61 ± 0.12**	0.96 ± 0.03	1	0.46 ± 0.02**	0.89 ± 0.01**

The data are based on at least three individual measurements in each plant species. In each functional group, the data in the -NO_2_ group is taken as the baseline data, and those in the corresponding +NO_2_ and NR groups are presented as the mean ± standard errors of the folds compared to the corresponding baseline data. One-way analysis of variance after Levene’s test was performed for statistical analysis. In case of heterogeneity of variance, nonparametric Kruskal-Wallis one-factor analysis of variance was used. * *p* < 0.05 and ** *p* < 0.01 compared to the corresponding -NO_2_ group in the same functional group.

**Table 3 plants-08-00045-t003:** Relative Chl content, peroxidase (POD) activity, soluble protein concentration, and malondialdehyde (MDA) concentration (folds) in different plant functional groups treated without nitrogen dioxide (NO_2_) (−NO_2_), with 72 h NO_2_ stress (+NO_2_) or after 30 d of natural recovery (NR).

	Relative Chl Content			Relative POD Activity			Relative Soluble Protein Concentration			Relative MDA Concentration		
	−NO_2_	+NO_2_	NR	−NO_2_	+NO_2_	NR	−NO_2_	+NO_2_	NR	−NO_2_	+NO_2_	NR
**Life from**												
Herb	1	1.23 ± 0.91	1.35 ± 1.02	1	6.14 ± 0.79**	1.22 ± 0.70	1	0.81 ± 0.41	0.68 ± 0.38**	1	1.11 ± 0.61	0.71 ± 0.39**
Shrub	1	0.67 ± 0.30**	1.13 ± 0.76**	1	3.59 ± 0.87**	2.08 ± 0.33*	1	1.05 ± 0.54	0.77 ± 0.43*	1	4.84 ± 9.70*	2.60 ± 3.13
Tree	1	0.94 ± 0.50*	1.51 ± 0.84	1	1.18 ± 0.39	0.88 ± 0.51	1	1.41 ± 1.03	0.71 ± 0.45*	1	1.57 ± 0.98	0.81 ± 0.45
Evergreen	1	0.81 ± 0.43	1.76 ± 0.97	1	0.91 ± 0.63	0.62 ± 0.48*	1	0.78 ± 0.36*	0.35 ± 0.34**	1	1.07 ± 0.65	0.57 ± 0.39
Deciduous	1	1.07 ± 0.55	1.25 ± 0.61	1	1.45 ± 0.72	1.13 ± 0.40	1	2.01 ± 1.12**	1.06 ± 0.16	1	2.07 ± 1.00*	1.05 ± 0.37
Broadleaf	1	0.81 ± 0.48**	1.22 ± 0.78**	1	3.65 ± 0.57**	1.64 ± 0.90	1	1.09 ± 0.70	0.76 ± 0.42**	1	3.51 ± 7.68	1.86 ± 2.58
Needle-like	1	0.77 ± 0.24	2.32 ± 1.35	1	1.65 ± 0.07**	1.13 ± 0.66	1	0.84 ± 0.50	0.54 ± 0.09**	1	3.04 ± 0.77**	0.65 ± 0.45
**Phylogeny**												
Gymnosperm	1	0.68 ± 0.18**	1.59 ± 1.19	1	1.81 ± 0.21**	1.06 ± 0.25	1	1.37 ± 0.23*	0.62 ± 0.20*	1	2.44 ± 0.63**	1.04 ± 0.52
Angiosperm	1	0.82 ± 0.49**	1.24 ± 0.79*	1	3.74 ± 0.67**	1.68 ± 0.95	1	1.65 ± 0.69	0.75 ± 0.42**	1	3.60 ± 7.87	1.88 ± 2.65
Monocotyledon	1	0.97 ± 0.73	1.17 ± 0.95*	1	6.34 ± 0.26**	1.22 ± 0.75	1	0.84 ± 0.32	0.76 ± 0.31*	1	1.03 ± 0.62	0.76 ± 0.39
Dicotyledon	1	0.79 ± 0.41**	1.26 ± 0.76	1	3.31 ± 0.63**	1.78 ± 0.12*	1	1.09 ± 0.74	0.74 ± 0.45**	1	4.19 ± 8.64*	2.14 ± 2.87
**Photosynthetic pathway**												
C_4_ herb	1	1.05 ± 0.77	1.54 ± 1.13	1	10.00 ± 0.80**	1.41 ± 0.97	1	0.92 ± 0.24	0.77 ± 0.28	1	0.87 ± 0.25	0.69 ± 0.39
C_3_ herb	1	1.08 ± 0.72	1.17 ± 0.90	1	2.28 ± 0.79*	1.04 ± 0.11	1	0.62 ± 0.41	0.58 ± 0.45*	1	1.34 ± 0.77	0.72 ± 0.39
All species	1	0.81 ± 0.47**	1.28 ± 0.84*	1	3.74 ± 0.68**	1.68 ± 0.95	1	1.07 ± 0.70**	0.73 ± 0.42	1	3.48 ± 7.49**	1.80 ± 2.53

The data are based on at least three individual measurements in each plant species. In each functional group, the data in the −NO_2_ group is taken as the baseline data, and those in the corresponding +NO_2_ and NR groups are presented as the mean ± standard errors of the folds compared to the corresponding baseline data. One-way analysis of variance after Levene’s test was performed for statistical analysis. In case of heterogeneity of variance, nonparametric Kruskal-Wallis one-factor analysis of variance was used. * *p* < 0.05 and ** *p* < 0.01 compared to the corresponding -NO_2_ group in the same functional group.

**Table 4 plants-08-00045-t004:** Relative N, P, K, and Mg contents (folds) in different plant functional groups treated without nitrogen dioxide (NO_2_) (−NO_2_), with 72 h NO_2_ stress (+NO_2_) or after 30 d of natural recovery (NR).

	Relative N Content			Relative P Content			Relative K Content			Relative Mg Content		
	–NO_2_	+NO_2_	NR	–NO_2_	+NO_2_	NR	–NO_2_	+NO_2_	NR	–NO_2_	+NO_2_	NR
**Life from**												
Herb	1	1.30 ± 1.16	1.30 ± 0.80	1	0.95 ± 0.51**	1.33 ± 0.7	1	1.56 ± 0.57**	1.99 ± 0.74**	1	1.33 ± 0.42**	0.68 ± 0.45*
Shrub	1	1.13 ± 0.47	1.38 ± 1.17	1	0.74 ± 0.40**	0.97 ± 0.27**	1	1.17 ± 0.31	1.27 ± 0.16	1	1.25 ± 1.29*	1.51 ± 3.18**
Tree	1	1.19 ± 0.46	1.45 ± 1.01	1	1.22 ± 0.92	1.64 ± 0.94	1	1.28 ± 0.71	1.44 ± 0.58*	1	1.82 ± 1.17**	1.79 ± 2.05
Evergreen	1	1.20 ± 0.41	1.31 ± 0.33*	1	1.01 ± 0.45	1.87 ± 1.4	1	1.53 ± 0.79*	1.94 ± 0.74**	1	2.03 ± 1.50	2.48 ± 2.62
Deciduous	1	1.07 ± 0.49*	1.05 ± 0.33	1	1.44 ± 1.21	1.41 ± 0.65	1	1.42 ± 0.62**	1.35 ± 0.35*	1	1.61 ± 0.71**	1.09 ± 0.92
Broadleaf	1	1.19 ± 0.29	1.41 ± 1.07	1	0.89 ± 0.63**	1.17 ± 0.62*	1	1.39 ± 0.32*	1.87 ± 0.50**	1	1.44 ± 1.21*	1.51 ± 2.69**
Needle-like	1	1.10 ± 0.26	0.79 ± 0.59	1	1.2 ± 0.77	1.76 ± 1.00	1	0.81 ± 0.24	1.23 ± 0.7	1	1.60 ± 1.01	1.26 ± 0.46
**Phylogeny**												
Gymnosperm	1	0.97 ± 0.25	0.88 ± 0.41	1	1.06 ± 0.42	1.4 ± 0.79	1	1.17 ± 0.41	1.21 ± 0.47	1	1.05 ± 0.89*	1.48 ± 0.93
Angiosperm	1	1.52 ± 1.19	1.40 ± 1.08	1	0.86 ± 0.62**	1.17 ± 0.63**	1	1.63 ± 1.87	1.83 ± 1.95**	1	1.39 ± 1.22	1.5 ± 2.75**
Monocotyledon	1	1.34 ± 1.24	1.30 ± 0.86	1	0.8 ± 0.34**	1.16 ± 0.56	1	1.51 ± 0.59**	1.83 ± 0.65**	1	1.36 ± 0.44**	0.57 ± 0.37**
Dicotyledon	1	1.57 ± 1.18	1.42 ± 1.13	1	0.88 ± 0.7**	1.17 ± 0.65*	1	1.66 ± 2.06	1.83 ± 2.15	1	1.4 ± 1.33	1.71 ± 3.01**
**Photosynthetic pathway**												
C_4_ herb	1	1.19 ± 0.59	1.03 ± 0.64	1	0.8 ± 0.44	1.35 ± 0.69	1	1.28 ± 0.4	2.08 ± 0.69**	1	1.55 ± 0.46	0.63 ± 0.44
C_3_ herb	1	1.20 ± 0.53	1.57 ± 0.88*	1	1.10 ± 0.54*	1.12 ± 0.75	1	1.84 ± 0.58**	1.89 ± 0.81**	1	1.10 ± 0.19	0.72 ± 0.47
All species	1	1.67 ± 0.24*	1.24 ± 0.65	1	0.88 ± 0.60**	1.19 ± 0.65*	1	1.59 ± 1.78*	1.77 ± 1.87**	1	1.45 ± 1.20	1.49 ± 2.62**

The data are based on at least three individual measurements in each plant species. In each functional group, the data in the –NO_2_ group is taken as the baseline data, and those in the corresponding +NO_2_ and NR groups are presented as the mean ± standard errors of the folds compared to the corresponding baseline data. One-way analysis of variance after Levene’s test was performed for statistical analysis. In case of heterogeneity of variance, nonparametric Kruskal-Wallis one-factor analysis of variance was used. * *p* < 0.05 and ** *p* < 0.01 compared to the corresponding –NO_2_ group in the same functional group.

**Table 5 plants-08-00045-t005:** Relative Ca, Mn, and Zn contents (folds) in different plant functional groups treated without nitrogen dioxide (NO_2_) (–NO_2_), with 72 h NO_2_ stress (+NO_2_) or after 30 d of natural recovery (NR).

	Relative Ca Content			Relative Mn Content			Relative Zn Content		
	–NO_2_	+NO_2_	NR	–NO_2_	+NO_2_	NR	–NO_2_	+NO_2_	NR
**Life from**									
Herb	1	1.16 ± 0.41	0.94 ± 026*	1	1.57 ± 1.38	0.95 ± 0.60	1	1.08 ± 0.29	1.07 ± 0.90*
Shrub	1	1.07 ± 0.39	1.07 ± 0.36	1	1.24 ± 1.76	1.44 ± 0.91	1	0.95 ± 0.32	0.95 ± 0.62**
Tree	1	1.03 ± 0.42	0.94 ± 0.25*	1	2.80 ± 5.53	0.77 ± 1.03**	1	0.83 ± 0.39**	0.94 ± 0.30
Evergreen	1	1.94 ± 0.54**	1.09 ± 0.24	1	1.32 ± 1.11	0.53 ± 0.46**	1	0.70 ± 0.18**	0.99 ± 0.16
Deciduous	1	0.93 ± 0.20	0.78 ± 0.16**	1	4.27 ± 7.58	1.01 ± 1.36*	1	0.95 ± 0.50	0.89 ± 0.39
Broadleaf	1	1.04 ± 0.35	1.00 ± 0.32	1	1.65 ± 3.19*	1.51 ± 1.91	1	0.96 ± 0.35**	0.98 ± 0.64**
Needle-like	1	1.35 ± 0.85	1.23 ± 0.34	1	2.64 ± 0.09	0.30 ± 0.12	1	0.71 ± 0.12**	0.92 ± 0.01
**Phylogeny**									
Gymnosperm	1	1.04 ± 0.66*	1.00 ± 0.34	1	1.79 ± 1.02	0.44 ± 0.32**	1	0.93 ± 0.34	0.91 ± 0.20
Angiosperm	1	1.06 ± 0.35	1.01 ± 0.32	1	1.68 ± 3.27*	1.56 ± 1.95	1	0.95 ± 0.34**	0.98 ± 0.65**
Monocotyledon	1	1.21 ± 0.41	1.00 ± 0.23	1	1.77 ± 1.30	2.57 ± 3.85	1	1.10 ± 0.31	1.00 ± 0.95**
Dicotyledon	1	1.03 ± 0.33	1.01 ± 0.34	1	1.66 ± 3.58*	1.33 ± 1.03	1	0.91 ± 0.34**	0.98 ± 0.57**
**Photosynthetic pathway**									
C_4_ herb	1	1.46 ± 0.38*	1.08 ± 0.26	1	1.73 ± 1.11	0.84 ± 0.39	1	1.06 ± 0.40	1.24 ± 1.20*
C_3_ herb	1	0.85 ± 0.08**	0.80 ± 0.16**	1	1.46 ± 1.52	1.07 ± 0.76	1	1.12 ± 0.15	0.90 ± 0.43
All species	1	1.06 ± 0.39	1.01 ± 0.32	1	1.69 ± 3.12	1.38 ± 2.12	1	0.95 ± 0.34**	0.98 ± 0.62**

The data are based on at least three individual measurements in each plant species. In each functional group, the data in the −NO_2_ group is taken as the baseline data, and those in the corresponding +NO_2_ and NR groups are presented as the mean ± standard errors of the folds compared to the corresponding baseline data. One-way analysis of variance after Levene’s test was performed for statistical analysis. In case of heterogeneity of variance, nonparametric Kruskal-Wallis one-factor analysis of variance was used. * *p* < 0.05 and ** *p* < 0.01 compared to the corresponding -NO_2_ group in the same functional group.

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
