# Peer review of "Effects of Nitrogen Dioxide on Biochemical Responses in 41 Garden Plants"

_plants, 2019, doi:10.3390/plants8020045_

Round 1
Reviewer 1 Report
Overall a strong paper that examines the effect of nitrogen dioxide on antioxidant and chlorophyll concentrations in plants. However, protein content is not the same as protein carbonylation which is the preferred way to measure oxidative damage. Another potential issue is that basic statistical distribution checks were not done, i.e. Levene’s test to determine if groups had equality of variances and Shapiro–Wilk test to check for normality of the data. These are imports checks to do before an ANOVA.
Line 32 Clarify “limitations”
Line 38 change “this process would” to “this process does”
Line 62-67 units in mg/m3 and ul/L are used, please change to ul/L
Line 70 “to discuss the variation in mineral ion content” is repetitive and should be removed.
Line 74 Is green species a technical term?
Line 89 What other soils types were encountered, these are not specified.
Line 139 “3000 r for 10 min” should be changed to g units. This should be done throughout the document.
Line 142 450 nm, 5323, and 600 nm were used. How was the quantity calculated, were these averaged? Also, why were three wavelengths used, 532 nm is the standard.
Line 151 This assay was done as overtime, but the time length was not specified. Also, how was activity calculated (slope, min/max?)
Line 153-135 Dissolved protein is an odd method to measure oxidative damage. Why was protein carbonylation not used? Please provide a citation to justify this measurement.
Line 197 (Figure 1) There are font differences (fw is in a serif font while the rest of the figure is sans serif), groups are not continuous, a line should not be connecting the groups (this implies there is a specific order). Legend should be -NO2, +NO2, and NR, there is no reason to shorten an already short label. This critique also applies to Figure 2.
Line 222 was should be changed to were.
Line 254 “preferred” is an incorrect word choice, accessible is better. ROS can reach lipids and proteins while DNA is more shielded by cellular components.
Line 257 Protein oxidation should be measured by protein carbonylation assay.
Line 267 An alternative explanation as to why NO2 increases POD activity is that since chlorophyll is reduced fewer ROS are produced from the electron transport change leakage. Since there is reduced ROS burden POD has greater activity.
Line 279-286 Please rewrite to improve flow and add citations.
Line 312 “POD activity, and dissolved protein and” should be changed to “POD activity, dissolved protein and”
Author Response
Reviewer 1:
Overall a strong paper that examines the effect of nitrogen dioxide on antioxidant and chlorophyll concentrations in plants. However, protein content is not the same as protein carbonylation which is the preferred way to measure oxidative damage. Another potential issue is that basic statistical distribution checks were not done, i.e. Levene’s test to determine if groups had equality of variances and Shapiro–Wilk test to check for normality of the data. These are imports checks to do before an ANOVA.
Response: Thank you for your comment, which we think very constructive and from which we have benefitted a lot.
Following your comments, we have re-performed data analysis, and we do find some problems in the statistical distribution of the data.
In the revised manuscript, we have made the following statistical changes:
First, considering that a large variety of plants species were involved in this study and that the measurement data did not have the consistent baseline, we have transformed the measurement data to folds between the experimental groups and the control group in the same functional group. We believe this change can better reflect the variations in the investigated indices in different treatment situations.
Second, we have performed nonparametric nonparametric Kruskal-Wallis one-factor analysis of variance in the case of heterogeneity of variances. We have described the current statistical analysis section as follows:
“Measurement data are transformed to folds between the measurements in the experimental groups and those in the corresponding –NO2 (normal control) group. They are presented as the means ± SD (n = 3) and processed with SPSS 24.0 (IBM; Armonk, NY, USA). Levene’s test was performed to determine the homogeneity of variances, followed by one-way analysis of variance (ANOVA) for comparisons among the different treatment groups in the same functional group. In case of heterogeneity, nonparametric Kruskal-Wallis one-factor analysis of variance was used and Bonferroni corrections were performed.”
Third, we have transformed the original figures to tables to better convey the statistical outcomes, which can be seen in the current revised manuscript.
Line 32 Clarify “limitations”
Response: We apologize for the unclear description in the original manuscript. Here, we mean “the standard level for NO2 pollution”. According to China, the standard level of NO2 pollution is 0.01875 µl/L. In the revised manuscript, we have modified the original description to “… and the NO2 level will continue to exceed the standard level of NO2 pollution”.
Line 38 change “this process would” to “this process does”
Response: Thank you for your comment. We have changed “this process would” to “this process does…”.
Line 62-67 units in mg/m3 and ul/L are used, please change to ul/L
Response: Thank you for your feedback. We have used “ul/L” to replace the original unit “mg/m3” in the revised manuscript.
Specifically, “1.7, 4, 8.5 and 18.8 mg/m3” have been changed into “0.85, 2, 4.25, and 9.4 µl/L”, “18.8 mg/m3” into “9.4 µl/L”, and “4 and 8.5 mg/m3” into “2 and 4.25 µl/L”.
Line 70 “to discuss the variation in mineral ion content” is repetitive and should be removed.
Response: Thank you for your feedback. We have removed the sentence “to discuss the variation in mineral ion content”.
Line 74 Is green species a technical term?
Response: Thank you for your feedback. In the revised manuscript, we have used “landscaping plants” to replace “green species”.
Line 89 What other soils types were encountered, these are not specified.
Response: Thank you for your feedback. In the revised manuscript, we have added some detailed information with regard to the soil, as follows:
“The soil contained alkali-hydrolyzable N, available P, available K, and organic matters at 47.05 mg/kg, 5.91 mg/kg, 145.93 mg/kg, and 10.35 g/kg, respectively, with a pH value of 6.51.”
Line 139 “3000 r for 10 min” should be changed to g units. This should be done throughout the document.
Response: Thank you for your feedback. We have changed “r” to “g” throughout the manuscript.
Line 142 450 nm, 532, and 600 nm were used. How was the quantity calculated, were these averaged? Also, why were three wavelengths used, 532 nm is the standard.
Response: Thank you for your feedback. The calculation of the MDA content was not based on the averages of the recordings at the three wavelengths but on the following formula:
C/µg/Ml/L=6.45 (A532-A600)-0.56A450,
where A450, A532 and A600 represent the absorbance at a wavelength of 450, 532 and 600 nm, respectively.
In addition, we have cited the following reference to justify the calculation method:
Zhang, Z. K., Huber, D. J., Qu, H. X., Yun, Z., Wang, H., Huang, Z. H.. (2015). Enzymatic browning and antioxidant activities in harvested litchi fruit as influenced by apple polyphenols. Food Chemistry, 171, 191–199.
Line 151 This assay was done as overtime, but the time length was not specified. Also, how was activity calculated (slope, min/max?)
Response: Thank you for your comment. In the revised manuscript, we have added the following information to clarify this point:
“… 1 ml of 0.05 mol/L guaiacol (the donor) and 1 ml of 2% H2O2 (the substrate) for 15-min reaction. The absorbance at 470 nm was measured, and the activity was expressed as the optical density at 470 nm per minute per gram of fw, based on the following formula:
POD activity [u/(g•min)] =(△A470×Vt)/(W×VS×0.01×t)
where △A470 is the variation of the absorbance within the reaction time period, W is the fresh weight of the sample (g), t is the reaction time (1 min), Vt is the total volume of enzyme extract (6 mL), and Vs is the amount of use of the extract liquid at the time of measurement (0.1 mL).”
Line 153-135 Dissolved protein is an odd method to measure oxidative damage. Why was protein carbonylation not used? Please provide a citation to justify this measurement.
Response: Thank you for your feedback.
We apologize for our carelessness in the original manuscript, in which we miswrote “soluble protein” for “dissolved protein”. In the revised manuscript, we have corrected the mistake throughout the manuscript. In addition, to justify the measurement method for oxidative damage used in this study, we have cited references, as follows:
Shen, H.F.; Zhao, B. Study on evaluation of heat tolerance and its physiological mechanisms in Rhododendroncultivars. Plant Physiol J 2018, 54, 335-345 (in Chinese).
Bian, W.J.; Bao, G.Z.; Qian, H.M.; Song, Z.W.; Qi, Z.M.; Zhang, M.Y.; Chen, W.W.; Dong, W.Y. Physiological response characteristics in Medicago sativa under freeze-thaw and deicing salt stress. Water Air Soil Pollut 2018, 229, 196.
Line 197 (Figure 1) There are font differences (fw is in a serif font while the rest of the figure is sans serif), groups are not continuous, a line should not be connecting the groups (this implies there is a specific order). Legend should be -NO2, +NO2, and NR, there is no reason to shorten an already short label. This critique also applies to Figure 2.
Response: Thank you for your careful review and your comment is very important. In the revised manuscript, we have re-analyzed the data. To present the analytical outcomes more clearly, we have used tables to replace the original figures. In addition, we have replaced “-N” and “+N” with “-NO2” and “+NO2”, respectively, throughout the manuscript.
Line 222 was should be changed to were.
Response: Thank you for your careful review. We have made the correction in the revised manuscript.
Line 254 “preferred” is an incorrect word choice, accessible is better. ROS can reach lipids and proteins while DNA is more shielded by cellular components.
Response: Thank you for your thoughtful comments. We have changed the original “preferred” to “accessible” in the revised manuscript.
Line 257 Protein oxidation should be measured by protein carbonylation assay.
Response: Thank you for your feedback. We apologize for our carelessness in the original manuscript, in which we miswrote “soluble protein” for “dissolved protein”. To investigate protein oxidation, measurement of soluble protein is also a commonly-applied method. In the revised manuscript, we have cited some references to support this approach. Also, please refer to our response to one of the previous questions.
Line 267 An alternative explanation as to why NO2 increases POD activity is that since chlorophyll is reduced fewer ROS are produced from the electron transport change leakage. Since there is reduced ROS burden POD has greater activity.
Response: Thank you your comment, which we think very reasonable. Indeed, this is an alternative explanation why NO2 increases POD activity. Enlightened by your comment, as well as to avoid misleading potential readers, we have weakened the strength of the original description and added this alternative explanation as follows:
“NO2 increased the activity of POD, which might be associated with increased ROS generation (alternatively, since NO2 reduces chlorophyll, fewer ROS are produced from the electron transport change leakage; the reduced ROS burden leads to greater activity of POD).”
Line 279-286 Please rewrite to improve flow and add citations.
Response: Thank you for your feedback. We have rewritten the sentences in question and added citations when necessary. The changes are as follows:
“POD plays a role both in cell wall toughening and in the production of secondary metabolites; these roles, as well as its concurrent oxidant and antioxidant capabilities, make POD an important factor in the integrated defense response of plants to NO2 stress [11]
The balance of plant ions reflects the degree of environmental stability; it is the premise of maintaining normal physiological activities inside plant cells. Adversities can affect the normal physiology of plants: They alter ion concentrations in plant organs and thus disrupt the dynamic balance among different ions. Although increased NO2 stimulates plant growth, exposure to a high concentration of NO2 within a short period reduces mineral ion concentrations in plant issues [37, 38]”
Newly added references in this part:
Kume, A.; Tsuboi, N.; Suzuki, T.S.M.; Chiwa, M.; Sakurai, K.N.N.; Horikoshi, T.; Sakugawa, H. Physiological characteristics of Japanese red pine, Pinus densiflora Sieb. et Zucc., in declined forests at Mt. Gokurakuji in Hiroshima Prefecture, Japan. Trees 2000, 14, 305–311.
Tiwari, S.; Agrawal, M.; Marshall, F.M. Evaluation of ambient air pollution impact on carrot plants at a suburban site using open top chambers. Environmental Monitoring and Assessment 2006, 119, 15–30.
Line 312 “POD activity, and dissolved protein and” should be changed to “POD activity, dissolved protein and”
Response: We have made the suggested change. Thank you.
Reviewer 2 Report
Dear Authors,
this article is of great interest both for researchers and for experts in the design sector, also in urban areas.
At the moment it is not acceptable for publication.
Below is a series of suggestions
TITLE: maybe physiological is not the right term. It would be more biochemical. Information on minerals is missing.
ABSTRACT: no conclusions are missing.
INTRODUCTION: Line 34_ does not start with an abbreviation.
M & M: - locate the experiment area and also enter the altitude.
- how many days does exposure to NO2 last?
- the statistical analyzes conducted are fine but the range should be extended. The data are represented badly both in the table and in the figures. Please enter the letters obtained from the one-way ANOVA to identify the significant differences. In the figures, there are no statistics specifications. I suggest using tables or histograms since they are not temporal dynamics. Furthermore, have the data been transformed? Are they homogeneous and normal? Please specify.
RESULTS: at the moment this section is only descriptive since the statistical differences are not very clear. I suggest making the tables and figures clearer and specifying only significant differences. Why are the differences between the various species not presented individually? The data could also be described with a PCA.
DISCUSSION: once the results have been better defined, this discussion should be extended with comments with similar studies. You have studied many species and therefore I believe that some new REF is available.
CONCLUSION: so which species would be better to use?
The captions of the tables and figures must be improved. They must be able to explain the tables and figures alone.
Author Response
Reviewer 2:
this article is of great interest both for researchers and for experts in the design sector, also in urban areas.
At the moment it is not acceptable for publication.
Below is a series of suggestions
TITLE: maybe physiological is not the right term. It would be more biochemical. Information on minerals is missing.
Response: Thank you for your suggestion. We have changed “physiological” to “biochemical” in the revised manuscript.
ABSTRACT: no conclusions are missing.
Response: Thank you for your feedback. In the revised manuscript, we have modified the conclusion of the Abstract section to the following:
“NO2 poses a pollution risk to plant systems, and antioxidant status plays an important role in plant protection against NO2-induced oxidative damage, which is particularly noticeable in evergreen plants. The results of this study may provide useful data for the selection of landscaping plants in NO2 polluted areas.”
INTRODUCTION: Line 34_ does not start with an abbreviation.
Response: Thank you for your feedback. We have made the suggested change.
M & M: - locate the experiment area and also enter the altitude.
Response: Thank you for your comment. In the revised manuscript, we have provided the missing information as follows:
“The experiment was conducted in the garden plant laboratory and greenhouse of Nanjing Forestry University in Nanjing, Jiangsu Province, China (lat. 32.07°N, long. 118.8°E).”
- how many days does exposure to NO2 last?
Response: Thank you for your comment. The fumigation experiment lasted 3 days at 6 h/d. We have provided the missing information in the revised manuscript.
- the statistical analyzes conducted are fine but the range should be extended. The data are represented badly both in the table and in the figures. Please enter the letters obtained from the one-way ANOVA to identify the significant differences. In the figures, there are no statistics specifications. I suggest using tables or histograms since they are not temporal dynamics. Furthermore, have the data been transformed? Are they homogeneous and normal? Please specify.
Response: Thank you for your constructive comment. Following your comment, we have re-performed statistical analysis.
Considering that a large variety of plants species were involved in this study and that the measurement data did not have the consistent baseline, we have transformed the measurement data to folds between the experimental groups and the control group in the same functional group. We believe this change can better reflect the variations in the investigated indices in different treatment situations.
Even though, part of the data were heterogenous. For data with heterogeneity of variances, we have supplemented nonparametric Kruskal-Wallis one-factor analysis of variance. Accordingly, we have changed the description regarding statistically analysis to the following:
“Measurement data are transformed to folds between the measurements in the experimental groups and those in the corresponding –NO2 (normal control) group. They are presented as the means ± SD (n = 3) and processed with SPSS 24.0 (IBM; Armonk, NY, USA). Levene’s test was performed to determine the homogeneity of variances, followed by one-way analysis of variance (ANOVA) for comparisons among the different treatment groups in the same functional group. In case of heterogeneity, nonparametric Kruskal-Wallis one-factor analysis of variance was used and Bonferroni corrections were performed.”
In addition, to better present the analytical outcomes, we have used tables to replace the original figures.
RESULTS: at the moment this section is only descriptive since the statistical differences are not very clear. I suggest making the tables and figures clearer and specifying only significant differences. Why are the differences between the various species not presented individually? The data could also be described with a PCA.
Response: Thank you for your comment. Following your comment, we have re-performed statistical analysis. Based on the statistical outcomes, we have modified the original description as follows:
“The values of the various leaf characteristics differed greatly between functional groups (Table 2). The smallest variation in the Chl a content was observed in evergreens (0.9±0.05 fold that before NO2 exposure; p < 0.05); the greatest, in conifers (0.31±0.11 fold that before NO2 exposure; p <0.01). The variations in Chl b content ranged from 0.94±0.09 fold in evergreens (p > 0.05) to 0.35±0.04 fold in conifers (p < 0.01). The variations in the mean carotenoid content ranged from 0.92±0.17 fold in evergreens (p > 0.05) to 0.24±0.02 fold in conifers (p < 0.01).
Under NO2 stress, a very significant decrease in Chl content occurred and the relative Chl content significantly decreased to 0.81±0.47 fold that of the control (p < 0.01). After recovery, this content greatly increased. The results are summarized in Table 3.”
“The relative POD activity increased to 3.74±0.68 folds that of the control (p < 0.01) in response to NO2 treatment but decreased to 1.68±0.95 folds after 30 d of recovery, and the relative soluble protein concentration increased to 1.07±0.70 folds that of the control (p < 0.01) in response to NO2 treatment but decreased to 0.73±0.42 fold after 30 d of recovery. The relative MDA concentration increased to 3.48±7.49 folds that of the control (p < 0.01) in response to NO2 treatment but decreased to 1.80±2.53 folds after 30 d of recovery. Interestingly, the relative POD activity and the soluble protein and MDA concentrations in evergreens subjected to the NO2 and NR treatments were noticeably lower than those in the other functional groups subjected to the same treatments.”
In this study, we selected 41 species of landscape plants of different functional groups for investigation, which are the typical plants used for landscaping in East China. East China, particularly Nanjing, shows noticeable seasonal variations. In winter, deciduous plants shed leaves and enter the dominant period, showing great changes in physiological activities. By applying NO2 stress to the typical 41 plant species of different functional groups, we meant to screen the functional groups with relatively strong NO2 resistance. The outcomes of this study showed that evergreen plants displayed the strongest NO2 resistance among the selected functional groups. This finding implies that more evergreen plants and reduced deciduous plants should be allocated in the planning of landscape configuration, and this point is of particular significance for areas with NO2 pollution. In this study, we investigated the biochemical responses of plants to NO2 exposure based on the classification of functional groups. Numerous scholars have also used this design. For example, Chen et al (2012) had an article published in Environmental Pollution, in which the investigation was based on the classification of functional groups. Apart from the outcomes directly obtained from the experiment, this design also has the virtue that the obtained outcomes regarding the physiological configuration of the selected plant species may also be applicable to those uninvolved. Therefore, the outcomes obtained based on this design may have higher application and reference values.
Related reference:
Chen, J.; Wu, F.H.; Liu, T.W.; Chen, L.; Xiao, Q.; Dong, X.J.; He, J.X.; Pei, Z.M.; Zheng, H.L. Emissions of nitric oxide from 79 plants species in response to simulated nitrogen deposition. Environ Pollut 2012, 160, 192-200.
DISCUSSION: once the results have been better defined, this discussion should be extended with comments with similar studies. You have studied many species and therefore I believe that some new REF is available.
Response: Thank you for your concern. Following your constructive comment, we have re-performed analysis of the data. However, the analysis remains based on the classification of different functional groups. Our explanations can be found in our response to the previous question.
However, your comment is very enlightening, and research in this direction, based on the outcomes of this study, will be very interesting.
CONCLUSION: so which species would be better to use?
Response: Thank you for your concern. Based on the outcomes of this study, we have supplemented the following conclusion:
“… compared with those in other functional groups, species in some functional groups, especially evergreens, exhibited higher leaf area and leaf N levels, …”
The captions of the tables and figures must be improved. They must be able to explain the tables and figures alone.
Response: Thank you for your feedback. In the revised manuscript, we have updated the titles of tables and tried any means to let the titles explain the tables alone. These changes can be found in the current manuscript.
Round 2
Reviewer 2 Report
Dear Authors,
the present version of the article is considerably better than the previous one. I only have a few small suggestions:
- Abstract: so suggest the evergreen plants in conditions of strong pollution? I would better specify your conclusions.
- Introduction: line 38 delete (NO2).
lines 65-66: enter the scientific names of the species and not the vulgar names.
- lines 69-72: this part should also be specified in M & M.
- Table 1: in addition to the species are also included the names of the cultivars. To specify.
- M & M: line 108: do you mean 14 hours of photoperiods?
- Discussion: comments and citations relating to similar studies already present in the literature are lacking.
Author Response
the present version of the article is considerably better than the previous one. I only have a few small suggestions:
- Abstract: so suggest the evergreen plants in conditions of strong pollution? I would better specify your conclusions.
Response: Thank you for your feedback. In the revised manuscript, we have further specified the conclusions as follows:
“In conditions of strong air pollution, more evergreen plants may be considered in landscape design, particularly in seasonal regions.”
- Introduction: line 38 delete (NO2).
Response: We have made the suggested change. Thank you.
lines 65-66: enter the scientific names of the species and not the vulgar names.
Response: Thank you for your feedback. We have replaced the original names of the species with scientific names in the revised manuscript.
- lines 69-72: this part should also be specified in M & M.
Response: Thank you for your comment. Indeed, this part is important to justify the use of NO2 at 6 µl/L in the experiment. Following your suggestion, we have specified it in the Materials and Methods as, follows:
“Considering that plants remained alive in the condition of NO2 below 4.25 μl/L [22] but died in 9.4 μl/L NO2 [9], the NO2 concentration was determined at 6 μl/L in this experiment.”
- Table 1: in addition to the species are also included the names of the cultivars. To specify.
Response: Thank you for your feedback. We have carefully checked the original Table 1 and provided the cultivar names when needed. For those without cultivar names, they were the original plant species.
- M & M: line 108: do you mean 14 hours of photoperiods?
Response: Thank you for your careful review and we apologize for our carelessness. In the revised manuscript, we have corrected the mistake.
- Discussion: comments and citations relating to similar studies already present in the literature are lacking.
Response: Thank you for your comment, which we believe very constructive. Following your comment, we have added discussion on some related studies that are already present in the reference list. The addition is as follows:
“Air pollutants cause damage to plants in two forms: chronic injury due to a long-term exposure to pollutants at a low concentration and acute injury as a consequence of a short-term fumigation with pollutants at a high concentration [15]. This study focused on the latter damage caused by a short-term fumigation with NO2 at 6 μl/L.”
“Zhang et al. [26] investigated the enzymatic browning and antioxidant activities in harvested litchi (Litchi chinensis Sonn.) fruit after exposure to apple polyphenols, and proposed that plants could produce reactive oxygen species (ROS) under salt stress and that plants had their own ROS scavenging systems (antioxidant systems). Our study showed that plants built their ROS scavenging systems by resorting to activity metabolism.”
“According to a similar study conducted by Fang et al. [11], NO2 exposure elevated the levels of lipid peroxidation and protein oxidation, accompanied by the introduction of antioxidant enzyme activities. The results of our study were basically consistent with those reported [11].”
